# GEO: GENERATIVE ENGINE OPTIMIZATION

## ABSTRACT

The advent of large language models (LLMs) has ushered in a new paradigm of search engines that use generative models to gather and summarize information to answer user queries. This emerging technology, which we formalize under the unified framework of generative engines (GEs), has the potential to generate accurate and personalized responses, and is rapidly replacing traditional search engines like Google and Bing. Generative Engines typically satisfy queries by synthesizing information from multiple sources and summarizing them with the help of LLMs. While this shift significantly improves *user* utility and *generative search engine* traffic, it results in a huge challenge for the third stakeholder – website and content creators. Given the black-box and fast-moving nature of generative engines, content creators have little to no control over *when* and *how* their content is displayed. With generative engines here to stay, the right tools should be provided to ensure that creator economy is not severely disadvantaged. To address this, we introduce GENERATIVE ENGINE OPTIMIZATION (GEO), a novel paradigm to aid content creators in improving their visibility. In this work, we propose several optimizations that can be applied to improve the visibility of content. To evaluate and compare different GEO methods, we propose a benchmark encompassing diverse user queries from multiple domains and settings, along with relevant sources needed to answer those queries. Through rigorous experiments on the proposed benchmark, we demonstrate different GEO methods involving well-designed textual enhancements, are capable of boosting source visibility by up to 40% in GE responses. We find several insights that aid content creators – for example, adding citations and quotations significantly improves visibility. We also discover that these optimizations are domain dependent, thus requiring a change in the nature of the optimization based on the source. Our work opens a new frontier in the field of information discovery systems, with profound implications for both developers of GEs and content creators.

## 1 INTRODUCTION

The invention of traditional search engines three decades ago marked a shift in the way information was accessed and disseminated across the globe. While these search engines were powerful and ushered in a host of applications like academic research and e-commerce, they were limited to providing a list of relevant websites to user queries. The recent success of large language models (LLMs) however has paved the way for better systems like BingChat, Google's SGE, and perplexity.ai that combine the strength of conventional search engines with the flexibility of generative models. We dub these new age systems generative engines (GE), because they not only *search* for information, but also *generate* multi-modal responses by synthesizing multiple sources. From a technical perspective, generative engines involve retrieving relevant documents from a database (such as the internet) and using large neural models to generate a response grounded on the sources, to ensure attribution and a way for the user to verify the information.

The usefulness of generative engines for both their developers and users is evident – users can access information faster and more accurately, while developers can craft precise and personalized responses, both to improve user satisfaction and revenue. However, generative engines put the third stakeholder – website and content creators – at a disadvantage. While users would click on the website in traditional search engines, generative engines remove the need to navigate to websites by directly provide the information. Furthermore, the black-box nature of generative engines makes

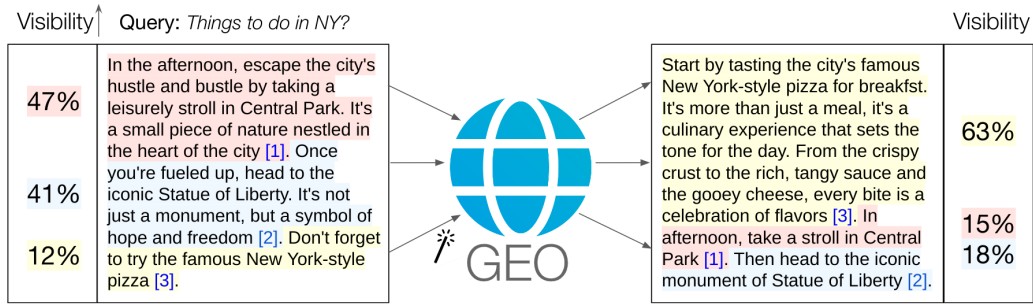

Figure 1: Our proposed GENERATIVE ENGINE OPTIMIZATIONmethod optimizes websites to boost their presence aka visibility in Generative Engine generated response. Unlike, search engines, where visibility is computed using ranking on the search engine page, Generative Engine requires specially designed metric to compute visibility. The figure shows, only Website 3 used GENERATIVE ENGINE OPTIMIZATION boosting it's visibility from 12% to 63%

it difficult for content creators to understand how their content is being used and displayed. With several individuals depending on the creator economy for their livelihood, in this work we introduce GENERATIVE ENGINE OPTIMIZATION (GEO), which is the first and important step towards providing them the tools to navigate this new technology.

GEO allows website owners to optimize their web content in terms of presentation, text style, and content. GENERATIVE ENGINE OPTIMIZATION methods can be thought of as optimization functions that take the source website as input and output an optimized version of the website, which has a higher likelihood of visibility in generative engines. In this paper, we develop various such methods and show their marked improvement in visibility in Generative Engines for a diverse set of input queries.

However, to measure visibility, a suitable metric is needed for generative engines. While average ranking on the search results page is a good measure of visibility in traditional search engines, defining visibility metrics for generative engines is non-trivial. This is because, generative engines provide a single text block with inline citations supporting statements of different sizes, positions, and presented in different ways. To this end, we propose suitable visibility metrics tailor-made for generative engines. These metrics measure visibility of attributed sources over multiple dimensions, such as relevance and influence of citation to query, through subjective and objective evaluations. They are designed to act as a standard that generative engine companies can provide to website owners to help them measure their site performance.

Through rigorous evaluations, we demonstrate that our proposed GENERATIVE ENGINE OPTIMIZATION methods can boost visibility by upto 40% on diverse set of queries, providing beneficial strategies for content creators to improve their visibility in the rapidly adapted generative engines. Among other things, we find that including citations, quotations from relevant sources, and statistics can significantly boost source visibility, with increase of over 40% across various queries. Further, we discover a dependence of the effectiveness of GENERATIVE ENGINE OPTIMIZATION methods on the domain of the query.

In summary, our contributions are four-fold: (1) We formalize the concept of Generative Engine and propose GENERATIVE ENGINE OPTIMIZATION as a new framework to allow website owners to start optimizing their content for the new search shift. (2) We propose specific visibility metrics that content creators can use to gauge their website's performance through subjective and objective evaluations. (3) We propose a new benchmark consisting of search queries from different domains and datasets specially repurposed for Generative Engines, with all queries categorized based on its type, domain, and other attributes. This benchmark serves as a starting point for evaluation in the new paradigm of Generative Engines. (4) We articulate different GEO strategies that website owners can use to significantly improve the visibility of their content by up to 40% with little effort. Through analysis, we also discuss different domain-specific strategies and the need for evolving traditional strategies for website optimization, as they have little to no effect in Generative Engines.

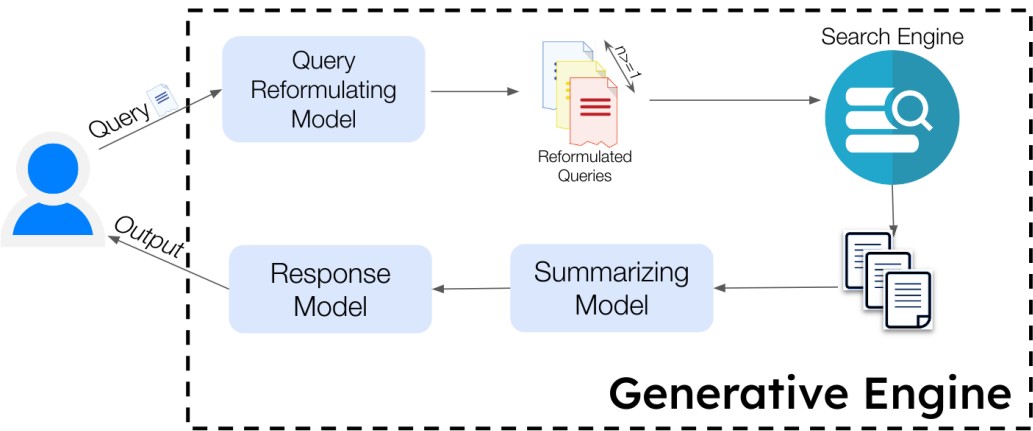

Figure 2: Overview of Generative Engines. Generative Engines primrarily consists of a set of generative models and a search engine to retrieve relevant documents. Generative Engines take user query as input and through a series of steps generate a final response that is grounded in the retrieved sources with inline attributions throught the response.

## 2  FORMULATION OF GENERATIVE ENGINES

Despite the deployment of a myriad of generative engines to millions of users already, there is currently no standard framework. We provide a formulation for that can accommodate various modular components incorporated in their design.

We describe a generative engine, which includes several backend generative models and a search engine for source retrieval. A Generative Engine (GE) takes as input a user query $q_u$ and returns a natural language response $r$, where $P_U$ represents personalized user information, such as preferences and history. The GE can be represented as a function:

$$f_{GE} := (q_u, P_U) \rightarrow r \tag{1}$$

While the response $r$ can be multimodal, we simplify it to a textual response in this section.

Generative Engines are comprised of two crucial components: a.) A set of generative models $G = \{G_1, G_2...G_n\}$, each serving a specific purpose like query reformulation or summarization, and b.) A search engine $SE$ that returns a set of sources $S = \{s_1, s_2...s_m\}$ given a query $q$. We present a representative workflow in Figure 2, which at the time of writing, closely resembles the design of BingChat. This workflow breaksdown the input query into a set of simpler queries that are easier to consume for the search engine. Given a query, a query re-formulator generative model, $G_1 = G_{qr}$, generates a set of queries $Q^1 = \{q_1, q_2...q_n\}$, which are then passed to the search engine $SE$ to retrieve a multi-set of ranked sources $S = \{s_1, s_2, ..., s_m\}$. The sets of sources $S$ are passed to a summarizing model $G_2 = G_{sum}$, which generates a summary $Sum_j$ for each source in $S$, resulting in the summary set ($Sum = \{Sum_1, Sum_2, ..., Sum_m\}$). The summary set is passed to a response generating model $G_3 = G_{resp}$, which generates a cumulative response $r$ backed by sources $S$.

The response $r$ is typically a structured text response along with citations embedded within the text to support the information provided. Citations are especially important given the tendency of LLMs to hallucinate information Ji et al. (2023). Specifically, consider a response $r$ composed of sentences $\{l_1, l_2...l_o\}$. Each sentence may be backed by a set of citations that are a part of the retrieved set of documents $C_i \subset S$. An ideal Generative Engine should ensure that all statements in the response are supported by relevant citations (high citation recall), and all citations accurately support the statements they're associated with (high citation precision) Liu et al. (2023a).

## 3  CONVERSATIONAL GENERATIVE ENGINE

In Section 2, we discussed a single-turn Generative Enginethat outputs a single response given the user query. However, one of the strengths of upcoming Generative Engines will be their ability

to engage in an active back-and-forth conversation with the user. The conversation allows users to provide clarifications to their queries or Generative Engineresponse and ask follow-ups. Specifically, in equation 1, instead of the input being a single query $q_u$, it is modeled as a conversation history $H = (q_u^t, r^t)$ pairs. The response $r^{t+1}$ is then defined as:

$$GE := f_{LE}(H, P_U) \rightarrow r^{t+1} \tag{2}$$

where $t$ is the turn number.

Further, to engage the user in a conversation, a separate LLM, $L_{follow}$ or $L_{resp}$, may generate suggested follow-up queries based on $H$, $P_U$, and $r^{t+1}$. The suggested follow-up queries are typically designed to maximize the likelihood of user engagement. This not only benefits Generative Engine providers by increasing user interaction, but also benefits website owners by enhancing their visibility. Furthermore, these follow-up queries can help users by getting more detailed information.

## 4    GENERATIVE ENGINE OPTIMIZATION

The advent of search engines led to the development of search engine optimization (SEO), a process to help website creators optimize their content to improve rankings in search engine results pages (SERP). Higher rankings correlate with higher visibility and increased website traffic. However, with generative engines becoming front-and-center in the information delivery paradigm and SEO not directly applicable to it, new techniques need to be developed.

To this end, we propose GENERATIVE ENGINE OPTIMIZATION, a new paradigm where content creators aim to increase their visibility in the generated responses. We define the visibility of a website/citation $c_i$ in a cited response $r$ from a generative engine by the function $g(c_i, r)$ and the website creator wants to maximize this. Simultaneously, from the perspective of the generative engine, the goal is to maximize the visibility of citations that are most relevant to the user query, i.e., maximize $\sum_i g(c_i, r) \cdot Rel(c_i, q, r)$, where $Rel(c_i, q, r)$ is a measure of the relevance of citation $c_i$ to the query $q$ in the context of response $r$. However, both the functions $g$ and $Rel$ are subjective and not well-defined yet for generative engines, and we define them below.

### 4.1    IMPRESSIONS FOR GENERATIVE ENGINES

In SEO, the impression (or visibility) of a website is simply determined by the average ranking of the website over a range of real queries. But given that the nature of output of generative engines is very different, impressions metrics are not yet defined. Unlike search engines, Generative Engines combine information from multiple sources in a single response. Thus multiple factors such as length, uniqueness and the presentation of the cited website determines the true visibility of a citation. In this section, we use website and citation interchangeably.

To address this, we propose several impression metrics. The "Word Count" metric is the normalized word count of sentences related to a citation. Mathematically, this is defined as:

$$Imp_{wc}(c_i, r) = \frac{\sum_{s \in S_{c_i}} |s|}{\sum_{s \in S_r} |s|} \tag{3}$$

Here $S_{c_i}$ is the set of sentences citing $c_i$, $S_r$ is the set of sentences in the response, and $|s|$ is the number of words in sentence $s$. In cases where a sentence is cited by multiple sources, we simply share the word count with the citations. Since "Word Count" is not impacted by the ranking of the citations (whether it appears first, for example), we propose a position-adjusted count which reduces the weight by an exponential decaying function of the rank of the citation:

$$Imp'_{wc}(c_i, r) = \frac{\sum_{s \in S_{c_i}} |s| \cdot e^{-\frac{pos(s)}{|S|}}}{\sum_{s \in S_r} |s|} \tag{4}$$

The above impression metrics are objective and well grounded. But they ignore the subjective aspects of the impact of citations on the user's attention. To address this, we propose the "Subjective Impression" metric, which incorporates multiple facets such as 1.) relevance of the cited material to the user query, 2.) influence of the citation, which evaluates the degree to which the generated

response depends on the citation, 3.) uniqueness of the material presented by a citation, 4.) subjective position, which measures how prominently the source is positioned from the user's perspective, 5.) subjective count, which measures the amount of content presented from the citation as perceived by the user upon reading the citation, 6.) probability of clicking the citation, and 7.) diversity in the material presented. In order to measure each of these sub-metrics, we use GPT-Eval Liu et al. (2023b), the current state-of-the-art for evaluation with LLMs which has a high correlation with human judgement for subjective tasks. We refer readers to Appendix B.3 for more details on the metric and its computation.

## 4.2 GENERATIVE ENGINE OPTIMIZATION METHODS FOR WEBSITE

To improve the impression metrics, content creators need to make changes to their websites. To this end, we present several generative engine-agnostic strategies, refered to as GENERATIVE ENGINE OPTIMIZATION methods (GEO). Mathematically, every GEO method is a function $f : W \rightarrow W_i'$, where $W$ is the initial web content, and $W'$ is the modified website content after applying LEO method. We propose and evaluate a series of methods.

**1: Authoritative:** Modifies text style of the source content to be more persuasive while making authoritative claims, **2. Statistics Addition:** Modifies content to include quantitative statistics instead of qualitative discussion, wherever possible, **3. Keyword Stuffing:** Modifies content to include more keywords from the query, as would be expected in classical SEO optimization. **4. Cite Sources & 5. Quotation Addition:** Adds relevant citations and quotations from credible sources, 6.) **6. Easy-to-Understand:** Simplifies the language of website, while **7. Fluency Optimization** improves the fluency of website text. **8. Unique Words & 9. Technical Terms:** involves adding unique and technical terms respectively wherever posssible,

In order to analyse the performance gain of our methods, for each input query, we randomly select one source to be optimized using all GEO separately. Further, for every method 5 answers are generated per query to reduce statistical noise in the results. We refer readers to Appendix B.4 for more details.

## 5 EXPERIMENTAL SETUP

### 5.1 EVALUATED GENERATIVE ENGINE

We use a 2-step setup for Generative Engine design: the first step involves fetching relevant sources for input query, followed by a LLM generating response based on the fetched sources. In our setup, we fetch top 5 sources from Google search engine for every query. The answer is generated by gpt3.5-turbo model using the prompt same as prior work Liu et al. (2023a). We refer readers to Appendix B for more details.

### 5.2 BENCHMARK

Since there is currently no publicly available dataset containing Generative Engine related queries, we curate **GEO-BENCH**, a benchmark consisting of 10K queries from multiple sources, repurposed for generative engines, along with synthetically generated queries. The benchmark includes queries from nine different sources, each further categorized based on their target domain, difficulty, query intent, and other dimensions.

The datasets used in constructing the benchmark are as follows:

**1. MS Macro, 2. ORCAS-1, and 3. Natural Questions:** Kwiatkowski et al. (2019); Alexander et al. (2022); Craswell et al. (2021) These datasets contain real anonymized user queries from Bing and Google Search Engines. These three collectively represent the common set of datasets that are used in search engine related research. However, Generative Engines will be posed with far more difficult and specific queries with intent of synthesizing answer from multiple sources instead of searching for them. To this end, we re-purpose several other publicly available datasets: **4. AllSouls:** This dataset contains essay questions from "All Souls College, Oxford University". The queries in this dataset require Generative Engines to perform appropriate reasoning to aggregate information from multiple sources. **5. LIMA:** contains challenging questions requiring Generative

Engines to not only aggregate information but also perform suitable reasoning to answer the question (eg: writing short poem, python code.). **6. Davinci-Debtate** Liu et al. (2023a) contains debate questions generated for testing Generative Engines. **7. Perplexity.ai Discover:** These queries are sourced from Perplexity.ai's Discover section, which is an updated list of trending queries on the platform. **8. ELI-5:** This dataset contains questions from the ELI5 subreddit, where users ask complex questions and expect answers in simple, layman terms. **9. GPT-4 Generated Queries:** To supplement diversity in query distribution, we prompt GPT-4 to generate queries ranging from various domains (eg: science, history) and based on query intent (eg: navigational, transactional) and based on difficulty and scope of generated response (eg: open-ended, fact-based)

Our benchmark contains 10K queries split into 8K,1K,1K train/val/test splits. Every query is tagged into multiple categories gauging various dimensions such as intent, difficulty, domain of query and format of answer type using GPT-4. We maintain the real-world query distribution, with our benchmark containing 80% informational queries, and 10% transactional and 10% navigational queries. We augment every query with cleaned text content of top 5 search results from Google search engine. We believe GEO-BENCH is a comprehensive benchmark for evaluating Generative Engines and serves as a standard testbed for evaluating Generative Engines for multiple purposes in this and future works. More details can be found in Appendix B.2.

## 5.3 EVALUATION METRICS

We evaluate all methods by calculating the Relative Improvement in Impression. For an initial generated Response $r$ from sources $S_i \in \{s_1, \ldots, s_m\}$, and a modified response $r'$, the relative improvement in impression of each source $s_i$ is measured as:

$$Improvement_{s_i} = \frac{Imp_{s_i}(r') - Imp_{s_i}(r)}{Imp_{s_i}(r)} * 100 \qquad (5)$$

## 6 RESULTS

Table 1: Performance improvement of GEO methods on GEO-BENCH. Performance Measured on Two metrics and their sub-metrics. Compared to the baselines simple methods such as Keyword Stuffing traditionally used in SEO do not perform very well. However, our proposed methods such as Statistics Addition and Quotation Addition show strong performance improvements across all metrics considered. The best performing methods improve upon baseline by 37% and 29% on Position-Adjusted Word Count and Subjective Impression respectively. For readability, Subjective Impression scores are normalized with respect to Position-Adjusted Word Count resulting in baseline scores being similar across the metrics

| Method | Position-Adjusted Word Count | | | Subjective Impression | | | | | | | |
| --- | --- | --- | --- | --- | --- | --- | --- | --- | --- | --- | --- |
| | Word | Position | **Overall** | Rel. | Infl. | Unique | Div. | FollowUp | Pos. | Count | **Average** |
| Performance without GENERATIVE ENGINE OPTIMIZATION | | | | | | | | | | | |
| **No Optimization** | 19.7 | 19.6 | 19.8 | 19.8 | 19.8 | 19.8 | 19.8 | 19.8 | 19.8 | 19.8 | 19.8 |
| Non-Performing GENERATIVE ENGINE OPTIMIZATION methods | | | | | | | | | | | |
| **Keyword Stuffing** | 19.6 | 19.5 | 19.8 | 20.8 | 19.8 | 20.4 | 20.6 | 19.9 | 21.1 | 21.0 | 20.6 |
| **Unique Words** | 20.6 | 20.5 | 20.7 | 20.8 | 20.3 | 20.5 | 20.9 | 20.4 | 21.5 | 21.2 | 20.9 |
| High-Performing GENERATIVE ENGINE OPTIMIZATION methods | | | | | | | | | | | |
| **Easy-to-Understand** | 21.5 | 22.0 | 21.5 | 21.0 | 21.1 | 21.2 | 20.9 | 20.6 | 21.9 | 21.4 | 21.3 |
| **Authoritative** | 21.3 | 21.2 | 21.1 | 22.3 | 22.9 | 22.1 | 23.2 | 21.9 | 23.9 | 23.0 | 23.1 |
| **Technical Terms** | 22.5 | 22.4 | 22.5 | 21.2 | 21.8 | 20.5 | 21.1 | 20.5 | 22.1 | 21.2 | 21.4 |
| **Fluency Optimization** | 24.4 | 24.4 | 24.4 | 21.3 | 23.2 | 21.2 | 21.4 | 20.8 | 23.2 | 21.5 | 22.1 |
| **Cite Sources** | 25.5 | 25.3 | 25.3 | 22.8 | 24.2 | 21.7 | 22.3 | 21.3 | 23.5 | 21.7 | 22.9 |
| **Quotation Addition** | 27.5 | 27.6 | **27.1** | 24.4 | 26.7 | 24.6 | 24.9 | 23.2 | 26.4 | 24.1 | **25.5** |
| **Statistics Addition** | 25.8 | 26.0 | 25.5 | 23.1 | 26.1 | 23.6 | 24.5 | 22.4 | 26.1 | 23.8 | 24.8 |

We evaluate a variety of GENERATIVE ENGINE OPTIMIZATION methods, each designed to optimize website content for better visibility in Generative Engine responses. These methods are compared against a baseline scenario where no optimization was applied. Our evaluation was conducted on GEO-BENCH, a diverse benchmark encompassing a wide array of user queries from multiple domains and settings. The performance of these methods was measured using two distinct metrics:

*Position-Adjusted Word Count* and *Subjective Impression*. The *Position-Adjusted Word Count* metric considers both the word count and the position of the citation in the GE's response, while the *Subjective Impression* metric incorporates multiple subjective factors to compute an overall impression score.

Our results, detailed in Table 1, reveal that our GENERATIVE ENGINE OPTIMIZATION methods consistently outperform the baseline across all metrics when evaluated on GEO-BENCH. This demonstrates the robustness of these methods to varying queries, as they were able to yield significant improvements despite the diversity of the queries. Specifically, our top-performing methods, namely Cite Sources, Quotation Addition, and Statistics Addition, achieved a relative improvement of 30-40% on the *Position-Adjusted Word Count* metric and 25-35% on the *Subjective Impression* metric compared to the baseline.

These methods, which involve adding relevant statistics (Statistics Addition), incorporating credible quotes (Quotation Addition), and including citations from reliable sources (Cite Sources) in the website content, require minimal changes to the actual content itself. Yet, they significantly improve the website's visibility in Generative Engine responses, enhancing both the credibility and richness of the content.

Interestingly, stylistic changes such as improving the fluency and readability of the source text, i.e methods Fluency Optimization and Easy-to-Understand also resulted in a significant boost of 10-20% in visibility. This suggests that Generative Engines not only value the content but also the presentation of the information.

Further, given generative models used in Generative Engine often are designed to follow instructions, one would expect a more persuasive and authoritative tone in website content can boost visibility. However, to the contary we find no significant improvement, demonstrating that Generative Engines are already somewhat robust to such changes. This points towards need for website owners to focus more towards improving presentation of content and making it more credible.

Finally, we also evaluate the idea of using keyword stuffing, i.e adding more relevant keywords in the website content. While this technique has been widely used for Search Engine Optimization, we find such methods have little to no performance improvement on Generative Engine's responses. This underscores the need for website owners to rethink their optimization strategies for Generative Engines, as techniques effective for traditional SEO may not necessarily translate to success in the new paradigm.

## 7 ANALYSIS

Table 2: Top Performing categories for each of the GENERATIVE ENGINE OPTIMIZATION methods. Website-owners can choose relevant GEO strategy based on their target domain.

Table 3: Visibility changes through GEO methods for sources with different Rankings in Search Engine. GEO methods are specially helpful for websites ranked lower in Search Engine rankings.

| Method | Top Performing Tags | | | Method | Relative Improvement (%) in Visibility | | | | |
|---|---|---|---|---|---|---|---|---|---|
| | Rank-1 | Rank-2 | Rank-3 | | Rank-1 | Rank-2 | Rank-3 | Rank-4 | Rank-5 |
| **Authoritative** | Debate | History | Science | **Authoritative** | -6.0 | 4.1 | -0.6 | 12.6 | 6.1 |
| **Fluency Opt.** | Business | Science | Health | **Fluency Opt.** | -2.0 | 5.2 | 3.6 | -4.4 | 2.2 |
| **Cite Sources** | Statement | Facts | Law & Gov. | **Cite Sources** | -30.3 | 2.5 | 20.4 | 15.5 | 115.1 |
| **Quotation Addition** | People & Society | Explanation | History | **Quotation Addition** | -22.9 | -7.0 | 3.5 | 25.1 | 99.7 |
| **Statistics Addition** | Law & Gov. | Debate | Opinion | **Statistics Addition** | -20.6 | -3.9 | 8.1 | 10.0 | 97.9 |

### 7.1 DOMAIN-SPECIFIC GENERATIVE ENGINE OPTIMIZATIONS

In Section 6, we presented the improvements achieved by GENERATIVE ENGINE OPTIMIZATION across the entirety of the GEO-BENCH benchmark. However, it is important to note that in real-world SEO scenarios, domain-specific optimizations are often applied to websites. With this in mind, and considering that we provide categories for every query in GEO-BENCH, we delve deeper into the performance of various GEO methods across these categories.

Table 7 provides a detailed breakdown of the categories where our GEO methods have proven to be most effective. A careful analysis of these results reveals several intriguing observations. For

instance, Authoritative significantly improves performance in the context of debate-style questions and queries related to the "historical" domain. This observation aligns with our intuition, as more persuasive form of writing is likely to hold more value in debates like contexts.

Similarly, the addition of citations through Cite Sources is particularly beneficial for factual questions. This is likely because citations provide a source of verification for the facts presented, thereby enhancing the credibility of the response. The effectiveness of different GEO methods varies across different domains. For example, as shown in row 5 of Table 7, domains such as 'Law & Government' and question types like 'Opinion' benefit significantly from the addition of relevant statistics in the website content, as implemented by Statistics Addition. This suggests that the incorporation of data-driven evidence can enhance the visibility of a website in particular contexts specially these. The method Quotation Addition is most effective in the 'People & Society', 'Explanation', and 'History' domains. This could be because these domains often involve personal narratives or historical events, where direct quotes can add authenticity and depth to the content.

Overall, our analysis suggests that website owners should strive towards making domain-specific targeted adjustments to their websites for higher visibility.

## 7.2 SIMULTANEOUS OPTIMIZATION OF MULTIPLE WEBSITES

In the evolving landscape of Generative Engines, it is anticipated that GEO methods will be widely adopted, leading to a scenario where all source contents are optimized using GEO. To understand the implications of this scenario, we conducted an evaluation of GENERATIVE ENGINE OPTIMIZATION methods by optimizing all source contents simultaneously. The results of this evaluation are presented in Table 7. A key observation from our analysis is the differential impact of GEO on websites based on their ranking in the Search Engine Results Pages (SERP). Interestingly, websites that are ranked lower in SERP, which typically struggle to gain visibility, benefit significantly more from GEO than those ranked higher. This is evident from the relative improvements in visibility shown in Table 7. For instance, the Cite Sources method led to a substantial 115.1% increase in visibility for websites ranked fifth in SERP, while on average the visibility of the top-ranked website decreased by 30.3%.

This finding underscores the potential of GEO as a tool to democratize the digital space. Importantly, many of these lower-ranked websites are often created by small content creators or independent businesses, who traditionally struggle to compete with larger corporations that dominate the top rankings in search engine results. The advent of Generative Engines may initially seem disadvantageous to these smaller entities. However, the application of GEO methods presents an opportunity for these small content creators to significantly improve their visibility in Generative Engine responses. By enhancing their content using GEO, they can reach a wider audience, thereby leveling the playing field and allowing them to compete more effectively with larger corporations in the digital space.

## 7.3 QUALITATIVE ANALYSIS

We present a qualitative analysis of GEO methods in Table 4. The analysis contains representative exmaples, where GEO methods boost source visibility while making minimal changes. For each of the three methods, a source is optimized by making suitable additions and deletions in the text. In the first example, we see, simply adding the source of a statement in text, can significantly boost visibility in final answer, requiring minimal effort on content creator's part. The second example demonstrates, addition of relevant statistics wherever possible, ensures source visibility increasing in the final Generative Engine response. Finally, the third row suggests, that merely emphasizing parts of the text and using a more persuasive text style can also lead to decent improvements in visibility.

## 8 RELATED WORK

**Evidence-based Answer Generation** Previous works have used several techniques for generating answers backed by relevant sources. Nakano et al. (2021) trained GPT-3 model to navigate web-based enviornment through textual commands, to answer questions backed by sources. Similarly

Table 4: Representative examples of GEO methods optimizing source website. Additions are marked in green and Deletions in red. Without adding any substantial new information in the content, GEO methods are able to significantly increase the visibility of the source content.

| Method | GEO Optimization | Relative Improvement |
|---|---|---|
| **Cite Sources** | **Query:** What is the secret of Swiss chocolate

With per capita annual consumption averaging between 11 and 12 kilos, Swiss people rank among the top chocolate lovers in the world (According to a survey conducted by The International Chocolate Consumption Research Group [1]) | **132.4%** |
| **Statistics Addition** | **Query:** Should robots replace humans in the workforce?

**Source:** Not here, and not now — until recently. The big difference is that the robots have come not to destroy our lives, but to disrupt our work, with a staggering 70% increase in robotic involvement in the last decade. | **65.5%** |
| **Authoritative** | **Query:** Did the jacksonville jaguars ever make it to the superbowl?

**Source:** It is important to note that The Jaguars have never appeared made an appearance in the Super Bowl. However, They have achieved an impressive feat by securing 4 divisional titles to their name. , a testament to their prowess and determination. | **89.1%** |

other methods Shuster et al. (2022); Thoppilan et al. (2022); Menick et al. (2022) fetch relevant sources through search engine and use them to generate answers. Our work tries to unify all these methods, and provide a common benchmark for improving these systems in future.

**Retrieval-Augmented Language Models:** Several, recent works have tackled the issues of limited memory of language models by fetching relevant sources from a knowledge base to complete a task Asai et al. (2021); Mialon et al. (2023); Guu et al. (2020). However, Generative Engine needs to not only generate answer, but also provide attributions throughout the answer. Further, Generative Engine is not limited to a single modality of text in terms of both input and output. Further, the framework of Generative Engine not limited to fetching relevant sources, but instead comprises of multiple tasks such as query reformulation, source selection, and taking decisions on how and when to perform them.

**Search Engine Optimization:** In nearly past 25 years, tremendous amount of public and private research has been done in optimizing web content for search engines Ankalkoti (2017); Shahzad et al. (2020); Kumar et al. (2019) These methods are typically classified into On-Page SEO, which involves improving actual content of the website and optimizing user experience and accessibility, and Off-Page SEO, which involves improving the website's authority and reputation through link building and recognition. In contrast, GEO deals with a more complex enviornment involving multi-modality, conversational settings. Further, since GEO is optimized against a generative model that is not limited to simple keyword matching, traditional SEO based strategies will not be applicable to Generative Engine settings highlighting the need for GEO.

## 9 CONCLUSION

In this work, we formulate the new age search engines that we dub generative engines and propose GENERATIVE ENGINE OPTIMIZATION (GEO) to help put the power in the hands of content creators to optimize their content. We define impression metrics for generative engines and propose a benchmark encompassing diverse user queries from multiple domains and settings, along with relevant sources needed to answer those queries. We propose several ways to optimize content for generative engines and demonstrate that these methods are capable of boosting source visibility by up to 40% in generative engine responses. Among other things, we find that including citations, quotations from relevant sources, and statistics can significantly boost source visibility. Further, we discover a dependence of the effectiveness of GENERATIVE ENGINE OPTIMIZATION methods on the domain of the query. Our work serves as a first step towards understanding the impact of generative engines on the digital space and the role of GENERATIVE ENGINE OPTIMIZATION in this new age of search engines.

## ETHICAL CONSIDERATIONS AND REPRODUCIBILITY STATEMENT

In our study, we focus on enhancing the visibility of websites in generative engines. We do not directly interact with sensitive data or individuals. While the sources we retrieve from search engines may contain biased or inappropriate content, these are already publicly accessible, and our study neither amplifies nor endorses such content. We believe that our work is ethically sound as it primarily deals with publicly available information and aims to improve the user experience in generative engines.

Regarding reproducibility, we have made our code available to allow others to replicate our results. Our main experiments have been conducted with five different seeds to minimize potential statistical deviations.

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

## A  CONVERSATIONAL GENERATIVE ENGINE

In Section 2, we discussed a single-turn Generative Enginethat outputs a single response given the user query. However, one of the strengths of upcoming Generative Engines will be their ability

to engage in an active back-and-forth conversation with the user. The conversation allows users to provide clarifications to their queries or Generative Engine response and ask follow-ups. Specifically, in equation 1, instead of the input being a single query $q_u$, it is modeled as a conversation history $H = (q_u^t, r^t)$ pairs. The response $r^{t+1}$ is then defined as:

$$GE := f_{LE}(H, P_U) \rightarrow r^{t+1} \qquad (6)$$

where $t$ is the turn number.

Further, to engage the user in a conversation, a separate LLM, $L_{follow}$ or $L_{resp}$, may generate suggested follow-up queries based on $H$, $P_U$, and $r^{t+1}$. The suggested follow-up queries are typically designed to maximize the likelihood of user engagement. This not only benefits Generative Engine providers by increasing user interaction, but also benefits website owners by enhancing their visibility. Furthermore, these follow-up queries can help users by getting more detailed information.

## B EXPERIMENTAL SETUP

### B.1 EVALUATED GENERATIVE ENGINE

While the exact design specifications of popular Generative Engines are not public, based on previous works Liu et al. (2023a), and our experiments, most of them follow 2-step procedure for generating responses. The first step involves fetching relevant sources for input query, followed by a LLM generating response based on the fetched sources. In our setup, we fetch top 5 sources from Google search engine for every query. The answer is generated by gpt3.5-turbo model using a prompt same as prior work Liu et al. (2023a).

### B.2 BENCHMARK

Since, currently there is no publicly available dataset containing Generative Engine related queries we curate **GEO-BENCH**, a benchmark containing 10K queries from multiple sources repurposed for generative engines along with synthetically generated queries. The benchmark contains queries from nine different sources, each further categorized based on their target domain, difficulty, query intent and other dimensions. The datasets used in constructing the benchmark are: **1. MS Macro & 2. ORCAS-1:** contains real anonymized user queries from Bing Search Engine. and **3. Natural Questions:** containing queries from Google Search Engine. These three collectively represent the common set of datasets that are used in search engine related research. However, Generative Engines will be posed with far more difficult and specific queries with intent of synthesizing answer from multiple sources instead of search for them. To this end, we re-purpose several other publicly available datasets: **4. AllSouls:** A dataset containing essay questions from "All Souls College, Oxford University". The queries in the dataset cannot be usually answered from a single source, and requires Generative Engines to aggregate information multiple sources and perform reasonable reasoning on them. **5. LIMA** Zhou et al. (2023) contains carefully crafted queries and responses for training pretrained language models for instruction following. The queries in the dataset represent a more challenging distribution of queries asked in Generative Engine, and often requires LLM's creative and technical powress to generate answers (eg: writing short poem, python code.) **6. Perplexity.ai Discover:** These queries are sourced from Perplexity.ai's, a public Generative Engine, Discover section which is a updated list of trending queries on the platform. These queries represent a real distribution of queries made on Generative Engines. **7. Davinci-Debtate** Liu et al. (2023a) contains debate questions generated using text-davinci-003 and sourced from Perspectrum dataset Chen et al. (2019). This dataset were specifically designed for Generative Engines. **8. ELI-5:** contains questions from the ELI5 subreddit, where users ask complex questions and expect answers in simple, layman terms. **9. GPT-4 Generated Queries:** To further supplement diversity in query distribution and increase Generative Engine specific queries, we prompt GPT-4 to generate queries ranging from various domains (eg: science, history), based on query intent (eg: navigational, transactional), based on difficulty and scope of generated response (eg: open-ended, fact-based).

In total our benchmark contains 10K queries split into 8K,1K,1K train/val/test splits. Every query is tagged into multiple categories gauging various dimensions such as intent, difficulty, domain of query and format of answer type using GPT-4. In terms of query intent, we maintain the real-world

query distribution, with our benchmark containing 80% informational queries, and 10% transactional queries and 10% navigational queries Jansen et al. (2008). Further, we augment every query with cleaned text content of top 5 search results from Google search engine. Owing to specially designed high benchmark diversity, size, complexity and real-world nature, we believe GEO-BENCH is a comprehensive benchmark for evaluating Generative Engines and serves as a standard testbed for evaluating Generative Engines for multiple purposes in this and future works.

### B.3 EVALUATION METRICS

We evaluate all methods by measuring the Relative Improvement in Impression. Specifically, given an initial generated Response $R$ from sources $s_i$s, and a modified response $R'$ from sources $s_i'$s, we measure the relative improvement in impression of each of the source $s_i$ as:

$$Improvement_{s_i'} = \frac{Imp(R') - Imp(R)}{Imp(R)} * 100 \tag{7}$$

We use the impression metrics as defined in Section 4.1. Specifically, we use two impression metrics: **1. Position-Adjusted Word Count** which is a combination of word count and position count. To dissect effect of individual components, we also report individual scores on the 2 sub-metrics. **2. Subjective Impression** which is a subjective impression metric which is a combination of seven different aspects: Relevance of citation to query, influence of citaiton on response, diversity and uniqueness of information presented, likelihood of followup by the user, perceived rank and amount of information presented in the answer. All these sub-metrics are evaluated using GPT-3.5, using methodology similar to described as in G-Eval Liu et al. (2023b). However, since G-Eval scores are ill-calibrated, we need to suitably normalize them for fair and appropriate comparison. We normalize Subjective Impression scores with respect to baseline scores of Position-Adjusted Word Count to ensure same mean and standard deviation.

### B.4 GEO METHODS

To improve the impression metrics, content creators need to make changes to their websites. To this end, we present several generative engine-agnostic strategies, refered to as GENERATIVE ENGINE OPTIMIZATION methods (GEO). Mathematically, every GEO method is a function $f : W \rightarrow W_i'$, where $W$ is the initial web content, and $W'$ is the modified website content after applying LEO method. We propose and evaluate a series of methods.

**1: Authoritative:** Modifies text style of the source content to be more persuasive while making authoritative claims, **2. Statistics Addition:** Modifies content to include quantitative statistics instead of qualitative discussion, wherever possible, **3. Keyword Stuffing:** Modifies content to include more keywords from the query, as would be expected in classical SEO optimization. **4. Cite Sources & 5. Quotation Addition:** Adds relevant citations and quotations from credible sources, 6.) **6. Easy-to-Understand:** Simplifies the language of website, while **7. Fluency Optimization** improves the fluency of website text. **8. Unique Words & 9. Technical Terms:** involves adding unique and technical terms respectively wherever posssible,

In order to analyse the performance gain of our methods, for each input query, we randomly select one source to be optimized using all GEO separately. Further, for every method, 5 answers are generated per query to reduce statistical noise in the results.

## C RESULTS

We perform on 5 seeds, and present results in Table 5

| Method | Position-Adjusted Word Count | | | Subjective Impression | | | | | | | |
|---|---|---|---|---|---|---|---|---|---|---|---|
| | Word | Position | Overall | Rel. | Infl. | Unique | Div. | FollowUp | Pos. | Count | Average |
| Performance without GENERATIVE ENGINE OPTIMIZATION | | | | | | | | | | | |
| **No Optimization** | $19.7_{(\pm0.7)}$ | $19.6_{(\pm0.5)}$ | $19.8_{(\pm0.6)}$ | $19.8_{(\pm0.9)}$ | $19.8_{(\pm1.6)}$ | $19.8_{(\pm0.6)}$ | $19.8_{(\pm1.1)}$ | $19.8_{(\pm1.0)}$ | $19.8_{(\pm1.0)}$ | $19.8_{(\pm0.9)}$ | $19.8_{(\pm0.9)}$ |
| Non-Performing GENERATIVE ENGINE OPTIMIZATION methods | | | | | | | | | | | |
| **Keyword Stuffing** | $19.6_{(\pm0.5)}$ | $19.5_{(\pm0.6)}$ | $19.8_{(\pm0.5)}$ | $20.8_{(\pm0.8)}$ | $19.8_{(\pm1.0)}$ | $20.4_{(\pm0.5)}$ | $20.6_{(\pm0.9)}$ | $19.9_{(\pm0.9)}$ | $21.1_{(\pm1.0)}$ | $21.0_{(\pm0.9)}$ | $20.6_{(\pm0.7)}$ |
| **Unique Words** | $20.6_{(\pm0.6)}$ | $20.5_{(\pm0.7)}$ | $20.7_{(\pm0.5)}$ | $20.8_{(\pm0.7)}$ | $20.3_{(\pm1.3)}$ | $20.5_{(\pm0.3)}$ | $20.9_{(\pm0.3)}$ | $20.4_{(\pm0.7)}$ | $21.5_{(\pm0.6)}$ | $21.2_{(\pm0.4)}$ | $20.9_{(\pm0.4)}$ |
| High-Performing GENERATIVE ENGINE OPTIMIZATION methods | | | | | | | | | | | |
| **Easy-to-Understand** | $21.5_{(\pm0.7)}$ | $22.0_{(\pm0.8)}$ | $21.5_{(\pm0.6)}$ | $21.0_{(\pm1.1)}$ | $21.1_{(\pm1.8)}$ | $21.2_{(\pm0.9)}$ | $20.9_{(\pm1.1)}$ | $20.6_{(\pm1.0)}$ | $21.9_{(\pm1.1)}$ | $21.4_{(\pm0.9)}$ | $21.3_{(\pm1.0)}$ |
| **Authoritative** | $21.3_{(\pm0.7)}$ | $21.2_{(\pm0.9)}$ | $21.1_{(\pm0.8)}$ | $22.3_{(\pm0.8)}$ | $22.9_{(\pm0.8)}$ | $22.1_{(\pm0.9)}$ | $23.2_{(\pm0.7)}$ | $21.9_{(\pm0.4)}$ | $23.9_{(\pm1.2)}$ | $23.0_{(\pm1.1)}$ | $23.1_{(\pm0.7)}$ |
| **Technical Terms** | $22.5_{(\pm0.6)}$ | $22.4_{(\pm0.6)}$ | $22.5_{(\pm0.6)}$ | $21.2_{(\pm0.7)}$ | $21.8_{(\pm0.8)}$ | $20.5_{(\pm0.5)}$ | $21.1_{(\pm0.6)}$ | $20.5_{(\pm0.6)}$ | $22.1_{(\pm0.6)}$ | $21.2_{(\pm0.2)}$ | $21.4_{(\pm0.4)}$ |
| **Fluency Optimization** | $24.4_{(\pm0.8)}$ | $24.4_{(\pm0.6)}$ | $24.4_{(\pm0.8)}$ | $21.3_{(\pm0.9)}$ | $23.2_{(\pm1.5)}$ | $21.2_{(\pm1.0)}$ | $21.4_{(\pm1.4)}$ | $20.8_{(\pm1.3)}$ | $23.2_{(\pm1.8)}$ | $21.5_{(\pm1.3)}$ | $22.1_{(\pm1.2)}$ |
| **Cite Sources** | $25.5_{(\pm0.7)}$ | $25.3_{(\pm0.6)}$ | $25.3_{(\pm0.6)}$ | $22.8_{(\pm0.9)}$ | $24.2_{(\pm0.7)}$ | $21.7_{(\pm0.3)}$ | $22.3_{(\pm0.8)}$ | $21.3_{(\pm0.9)}$ | $23.5_{(\pm0.4)}$ | $21.7_{(\pm0.6)}$ | $22.9_{(\pm0.5)}$ |
| **Quotation Addition** | $27.5_{(\pm0.8)}$ | $27.6_{(\pm0.8)}$ | $27.1_{(\pm0.6)}$ | $24.4_{(\pm1.0)}$ | $26.7_{(\pm1.1)}$ | $24.6_{(\pm0.7)}$ | $24.9_{(\pm0.9)}$ | $23.2_{(\pm0.9)}$ | $26.4_{(\pm1.0)}$ | $24.1_{(\pm1.2)}$ | $25.5_{(\pm0.9)}$ |
| **Statistics Addition** | $25.8_{(\pm1.2)}$ | $26.0_{(\pm0.8)}$ | $25.5_{(\pm1.2)}$ | $23.1_{(\pm1.4)}$ | $26.1_{(\pm0.9)}$ | $23.6_{(\pm0.9)}$ | $24.5_{(\pm1.2)}$ | $22.4_{(\pm1.2)}$ | $26.1_{(\pm1.2)}$ | $23.8_{(\pm1.2)}$ | $24.8_{(\pm1.1)}$ |

Table 5: Performance improvement of GEO methods on GEO-BENCH. Performance Measured on Two metrics and their sub-metrics.

