# OpenReview forum: "GEO: Generative Engine Optimization"
_ICLR.cc/2024/Conference — ICLR 2024 Conference Withdrawn Submission_

### Official Review · Reviewer_PP6U · 2023-10-30

**Soundness:** 2 fair
**Presentation:** 2 fair
**Contribution:** 2 fair
**Rating:** 3
**Confidence:** 3

**Summary:**

This paper focuses on the visibility evaluation of the content in the new generative search eigen empowered by the recent success of generative AI. This paper proposes some metrics to evaluate the visibility of the contents by a generative search engine. Based on the proposed metrics, the authors propose several suggestions to the content creators to improve their websites to increase the visibility of their content. The authors also prepare a benchmark dataset to evaluate the performance of the optimization they proposed in this paper and provide some simple analysis of the results.

**Strengths:**

1. A benchmark dataset is provided.

2. Some evaluation metrics are proposed.

3. Some SEO suggestions are provided.

**Weaknesses:**

1. The two-step experiment setting does not convince me of the evaluation. In the experiments, several sources are fetched from Google search first (Top 5), then the generative SO generates responses to the query. Based on this two-step setting, the visibility of the contents highly depends on the results returned by the traditional search eigen.

2. The scope of this paper may not attract many researchers from the ICLR community since there are limited contributions to the learning and representation perspectives.

3. The suggestions provided in this paper are straightforward without much deep insight.

4. The related works are not well discussed. Also, the citation style in the paper is not professional.

**Questions:**

1. In the evaluation, after modifying the contents, do we re-run the whole experiments following a two steps settings? If so, how to determine the modifications contribute to which step?

2. In the evaluation, following the two steps setting, how to determine the citation of each source?

---

### Official Review · Reviewer_DAsT · 2023-11-02

**Soundness:** 3 good
**Presentation:** 2 fair
**Contribution:** 2 fair
**Rating:** 3
**Confidence:** 3

**Summary:**

This paper introduces a method to improve the visibility of websites in generative engines. The authors first compare the differences in visibility measurement methods between traditional search engines and generative engines and propose visibility measurement indicators suitable for generative engines. Next, the article introduces the specific methods of GEO. Experimental results verify the proposed GEO can significantly improve the visibility of websites in generative engines.

**Strengths:**

- This paper explores a new problem when LLMs are served as generative engines—content creators lose their control of how the content is displayed, because the direct information provided by the generative engines reduces the invisibility of original content, which leads to economic losses for content creators.
- This paper proposes Word Count metric and  Position-Adjusted Count metric for invisibility when using generative engines.
- This paper proposes several methods to alleviate the invisibility problem, such as authoritative, statistics addition, etc. Experiments demonstrate their effectiveness.

**Weaknesses:**

- The paper is not organized well. The main text should exist independently, and the appendices should only supplement the main text. However, most of the results (e.g., various tables) of the main text are placed in the appendix.
- This paper uses GPT-Eval to evaluate the subjective impression in Table 1. The difference of difference dimensions is trivial and even completely the same in the first case. It is doubted whether this measurement is really valid and sufficient to support the conclusions of this paper.
- The invisibility problem is not clarified well. Traditional search engines retrieve content and rank them. If a user clicks content, then traffic comes to the website. Traffic monetization brings money to content creators. For generative engines, what does the improvement of visibility help specifically? Does the improvement of word count help the economy of content creators?

**Questions:**

Please refer to Weaknesses.

**Details Of Ethics Concerns:**

No ethical issues.

---

### Official Review · Reviewer_Lvo2 · 2023-11-06

**Soundness:** 1 poor
**Presentation:** 2 fair
**Contribution:** 2 fair
**Rating:** 3
**Confidence:** 5

**Summary:**

This paper introduces 'Generative Engine Optimization' which talks about how website creators can optimize their websites for visibility in Generative search results. It also proposes various visibility metrics that content/website creators can use to gauge their website's performance.

**Strengths:**

The paper addresses a very novel problem for the website creators which is how they can optimize their content in order to rank higher in the generative search results

**Weaknesses:**

Some of the weaknesses of the paper are:
1. The methodology is not thorough enough, I see mentions of functions g, rel in Section 4, but then they are not defined anywhere else.
2. The mathematical equations are not well defined and/or are incomplete. Specifically, I don't really understand S_ci and where it is coming from - Is it the set of sentences citing c_i *within* the response or not? Also the mathematical intuition or logic for calculating Imp_wc(c_i,r) is not clear to me.
3. How is the benchmark dataset 'GEO-Bench' generated is not very clear to me - i get the queries are sourced from 9 different places, but do we also include responses in this benchmark dataset?

**Questions:**

1. How is the benchmark dataset 'GEO-Bench' generated is not very clear to me - i get the queries are sourced from 9 different places, but do we also include responses in this benchmark dataset?
2. Mathematical intuition behind the defined metrics
3. I'm not clear how the modified response r' is generated? Also what exactly are we trying to improve - the evaluation metric defines 'Improvement_si' as the relative improvement in impression of each source s_i -> what does this mean? What will we achieve by improving the impression of a source s_i if it is already present in the response r?